# 3D Hand Scanning Methodology for Determining Protective Glove Dimensional Allowances

**DOI:** 10.3390/ijerph20032645

**Published:** 2023-02-01

**Authors:** Joanna Szkudlarek, Bartłomiej Zagrodny, Sandra Zarychta, Xiaoxue Zhao

**Affiliations:** 1Department of Personal Protective Equipment, Central Institute for Labour Protection—National Research Institute, 48 Wierzbowa Street, 90-133 Lodz, Poland; 2Department of Automation, Biomechanics and Mechatronics, Lodz University of Technology, 1/15 Stefanowskiego Street, 90-924 Lodz, Poland; 3Division of Dynamics, Lodz University of Technology, 1/15 Stefanowskiego Street, 90-537 Lodz, Poland

**Keywords:** personal protective equipment, dimensional allowances, ergonomic design, occupational health and safety, 3D hand scanning

## Abstract

There are two types of dimensional allowance (inner and external) related to two distinct areas of occupational health and safety: those being a measure of fit of personal protective equipment (PPE) and those determining the safe and comfortable human interaction with tools and machines, e.g., the latter ones result from wearing PPE increasing the dimensions of the human body and generating limitations in the work environment. In this paper, they are taken to mean the difference between the dimensions of a bare and gloved hand (including glove construction and materials). Dimensional allowances are important in designing the work environment, e.g., machine control panels and tools. The absolute and relative maximum values of dimensional allowances determined in this study for a hand in a firefighter’s protective glove for the main anthropometric data are: 16.90 mm (5.90%) for length, 12.00 mm (13.77%) for width, and 15.70 mm (7.96%) for circumference. The obtained results are useful for designers, and especially for designing keys on control panels and LCD touch displays and monitors integrated with machines.

## 1. Introduction

Many research studies focus on hand anthropometric data providing us with the fundamental knowledge used mainly in the customization, medicine and ergonomic design fields. However, major works limit data only to one dimension [1,2]. Moreover, the existing methods of gathering hand anthropometric data do not seem to be adequate in terms of the complex anatomy and the high inaccuracy of taking measurements from different people [3,4]. Thus, three-dimensional anthropometric measurements attract many researchers as a good solution for hand anthropometric data acquisition in terms of accuracy, resolution, as well as efficiency [1,2,5,6].

In this case, 3D scanning seems to be a good solution, particularly in contrast to goniometric techniques. The use of 3D scanning to obtain anatomical parameters of the hand does not interfere with hand posture or movement as the analysis is conducted on the recorded images. However, acquiring a good-quality scans and corresponding hand model for anthropometric measurement and analysis seems to be more complicated. Furthermore, the process of 3D hand scanning is fraught with numerous problems concerning hand positioning and identifying hand position in a protective glove. The difficulty of hand scanning was already noted by Griffin and Ashdown [7,8], indicating the need for developing a repeatable 3D handscanning method to generate reliable data for product creation [7].

Hand posture assessment is a well-known challenging problem resulting from the complex anatomy and curvature of the human hand [3,9,10]. These, in turn, can be caused by many aspects, such as repositioning, muscle tension and hand tremors [11]. Therefore, the newly created hand anthropometric conditions cause difficulty in the assessment of its posture when we are able to make a re-scan. Furthermore, during the 3D hand scanning process in the glove, the identification of the position of the hand becomes very troublesome. It is worth noting, that in comparison to the whole body, the hand is much smaller, which means fewer pixels in captured images and more noisy data [11].

Given the various problems identified in 3D hand scanning for measurement purposes, different methodologies have been proposed. Griffin et al. [7] present the analysis of the 3D hand in the glove fit with the use of pattern drafting and draping software. Pressure map function in drafted patterns draped around the hand scan provides to see the tight and loose fit areas. The method of joining 3D scan models of the bare hand and hand in a protective glove, their alignment and merger as a way of gaps validation existing between the hand surface and the inner side of the protective glove, as well as their fit, is mentioned in [8]. Inner dimensional allowances provide an accurate description of the glove fit to the hand. However, on the other side, external dimensional allowances appearing as a consequence of using protective gloves increase hand dimensions, generating more limitations connected with the flexibility of interacting with the outside environment. These restrictions are related to work performance, i.e., limited hand access to restricted areas, machines and equipment control, etc.

Three-dimensional analysis of a gloved hand in terms of dimensional allowances determining safe interaction with the work environment has been largely absent from scientific reports. Depending on the hazards against which the gloves are supposed to provide protection, they differ in design and dimensions. These, in turn, are specified for particular types of gloves and related to the normative requirements that must be met to obtain EU certification. Dimensional allowances refer to the typical shapes of access openings and confined spaces indicating the great significance of interactions between persons wearing personal protective equipment (PPE) and the work environment. The infrastructure of the working environment consists of workplaces, tools and machines. In this respect, the knowledge of dimensional allowances values resulting from the use of protective gloves is essential when designing ergonomic workplace control panels.

The novel approach to obtaining hand anthropometric data from both 2D and 3D scans is presented in [3]. The investigation includes two approaches to capturing 3D images. In the first method, 10 shots of volunteers’ plaster hands taken from different angles of view are transformed into one 3D image, whereas in the second method a combination of images from three different viewing angles is shown. In comparison to the first method, three 3D scanner shots do not provide complete hand surface data. The entire investigation seems to be very interesting providing many patterns that can be followed by other authors. Although in the mentioned article a dummy hand made of plaster was used it did not take into account the actual behavioral conditions imposed by the specific nature of the hand. Fingers segmentation as a tool for gathering information about the shape and dimensions of the hand can be also be applied, as in paper [12]. The authors present the analysis of the influence on the proportion of the fingers’ segments in different hand postures within the glove. The stretchable region was specified by additional holes in the glove. Then, repeated scans with affixing landmarks determining their position changes in different postures are measured from the 3D hand images. In [13], hand anthropometric analysis focused on hand postures as well as skin deformations with the use of a 3D scanner and is presented in relation to the improvement of glove fit, comfort and functionality. In the experiment, subjects with flat markers placed on their hands ware scanned a few times in each posture to complete a full 3D image.

Dimensional extraction can be conducted with various types of software [2,3,7,13]. A typical approach to evaluation of data from 3D image analysis consists of a comparison between the measurements taken from 3D scans with the ones gathered using tape or a caliper [3,12]. Paper [2], presents a validation of manual and 3D scan data for a resin dummy hand and an actual hand. Analisis of hand dimension is performed typically using means, standard deviations, root mean squared errors and correlation analysis calculations [2,3,12].

The purpose of this study is to obtain dimensioning results of a bare and gloved hand by using the customized method and approach. We present a much easier and more effective way to generate repeatable processes and reliable hand anthropometric data measurements involving 3D scanning. In terms of using the 3D scanning method, the extracted data indicate the distance between the hand and the external surface of the glove, which is essential for designing the working environment. In practice, the determination of dimensional allowances connected with the minimum space occupied by a worker’s hand in PPE and will have an immediate influence on the safety of manual work and the safety of operating with tools and machines. The importance of dimensional allowances, which are a consequence of the use of personal protective equipment (PPE) for work safety was discussed in a previous work [14]. In this paper, we propose a solution that improves the quality of the obtained data (dimensions) during the re-scanning process by combining positioning control with the other methods, such as marking landmarks to support dimensioning. The novelty is markers considered as landmarks determining the hand’s position in the glove are limited to the minimum and then used in the superimposing and dimensioning procedure.

## 2. Materials and Methods

The object of the research was a 25 year old, female volunteer. The subject did not have any hand anomalies, disorders or injuries. During the tests, the glove for firefighters was used, designed and fitted in compliance with the standard procedure [14,15]. This type of glove provides protection for firefighters against mechanical and thermal hazards [16,17,18,19]. Being made of Kevlar fibers and full-grain leather it also offers water resistance.

In this study, a professional and high-quality Artec Eva 3D handheld scanner (Artec Group, Luxembourg) with an accuracy of 0.1 mm and a resolution of 0.2 mm was used with the corresponding Artec Studio software for images processing [20]. For surface reconstruction of the obtained 3D scans, tools from MeshLab [21] software were applied, whereas for data extraction, CloudCompare [22] was used. The main advantage of the mentioned various software is the set of tools providing simple procedures for data measurements, as well as the overlapped cloud comparison. The key anthropometric data of the hand, such as the width, length and circumference, required by the standard ISO 21420:2003+A1: 2020 for protective gloves, aware included to describe the hand dimensions [23].

To provide the best quality of data acquisition from the 3D scanning procedure in the scanning and re-scanning process a palm rest was used to fix the hand and guarantee the same position, as shown in Figure 1a and Figure 2a. Such an arrangement eliminates the negative impact of involuntary hand movement from, e.g., muscle tension. On the hand itself and the hand in the protective glove, only a few landmarks limited only to the ones supporting the hand dimensioning were placed, providing the simplicity of taking measurements during the use of the software. Markers were designed and manufactured using 3D printing technology, made of resin, with a diameter of 4 mm. The location of the landmarks at key points (see Figure 1a and Figure 2a) determines the hand position in the glove, simplifying the data analysis process.

The preliminary research consisted of estimating the reliability of a 3D scanner and the variability of results in order to verify the 3D scan measuring method according to the assumptions. In this case, scans of the bare human hand concerning the reference points designated by markers were performed and compared with manualmeasurements taken with the use of a tape. The anthropometric data of the hand, such as length, width and circumference were extracted from 3D scans using the chosen software. The obtained results were analyzed. The positions of landmark on the bare hand during the 3D scanning procedure in real and with corresponding 3D scans are presented in Figure 1.

To determine the dimensional allowances between the bare and gloved hand, 15 scans for both were performed. On the glove, the supporting landmarks highlighting the hand position were placed. Based on the results, dimensional allowances defined as a difference between the dimensions of the bare hand and the hand in the protective glove were calculated. Minimum, maximum and mean dimensional allowances for length, width and circumference were calculated as crucial hand anthropometric data for the defined measuring points. Landmarks’ positions on the gloved hand during the 3D scanning procedure are presented in Figure 2 in real and with corresponding 3D scans.

In this study, two different approaches to measuring anthropometric hand dimensions were applied: linear dimensioning and superimposition. Length and width as linear dimensions were measured as distances between reference points, whereas for the circumference, CloudCompare software tools were applied. Prior to the dimensioning procedure, cleaning and surface reconstruction operations for each of the scanned objects were conducted in MeshLab software, mainly due to the errors caused by light reflections occurring during the 3D scanning procedure. The generated .stl files were uploaded to the CloudCompare software.

The *Cross Section* tool was used for the measurement of linear dimensions, such as length and width. It is represented by a clipping box with three principal axes (*x, y, z*), which can be easily extended or reoriented manually, according to the object being dimensioned, simultaneously displaying their values. For example, for the length, the bottom of the box is set to the landmark placed on the forearm, whereas its top reaches the tip of the third digit (see Table 1), giving an exact dimension.

For the calculation of circumference of the hand, firstly a point cloud of the object was generated. Then, a 1 mm high clipping box (*Cross Section* tool) was set to 20 mm above the thumb base (see Table 1). With the use of the *Export Envelope* button, information about a circumference was extracted, containing the measured dimension in the *Properties* Table.

For the calculation of t circumference dimensional allowances in the CloudCompare software, firstly the points clouds of the bare hand and hand in the protective glove were generated and superimposed. With the use of the *Cross Section* tool, a 1 mm hight measuring box was set at the appropriate level (see Table 1), to export new entities containing circumferences envelopes. The generated cross-sections were compared with the use of the *Compute cloud/Cloud distance* tool, providing us the information about the minimum, mean and maximum distances between bare and gloved hands.

The description of the manual and 3D scan measurements performed within mentioned stages is presented in Table 1. It contains the way of measuring anthropometric hand variables with the use of landmarks (reference points) for bare hand and hand in glove dimensioning has been presented in Figure 3.

To sum up, the CloudCompare procedure for measuring linear hand anthropometric data, such as length and width, is as follows:Upload an .stl file with a bare/gloved hand scan.Select the *Cross Section* tool.Adjust the dimensions of the clipping box using arrows according to the description of the hand variables (see Table 1) and landmark positions (see Figure 3).Read the data from the box.

For the measurement of the circumference:Upload the .stl file with the bare/gloved hand scan.Generate a point cloud of the object using the *Sample points on a mesh* tool.Select the *Cross Section* tool.Place a 1 mm high clipping box according to the description of the hand variable (see Table 1) and landmark positions (see Figure 3).Export envelope.Select a new entity and read the data from the *Properties* tab on the left side.

## 3. Results

The quality validation of results and 3D scanner accuracy have been analyzed for three hand parameters, i.e., length, width and circumference according to the defined measurements reference points. The subjects of comparison consisted of manual measurements and those made from scans (3D scan results). Mean and standard deviation values for each of the hand anthropometric parameters, in both methods, were calculated according to Formulas (1) and (2):(1)$$\bar{x}=\frac{1}{n}\overset{n}{\underset{i=1}{\sum }}x_{i}\;$$
where:

n—number of samples/size of the dataset,

i=1, 2, …, n—number of the current sample in the dataset,

$x_{i}$—the observed, current value of the sample in the dataset,

$\bar{x}$—mean of x values in the n-sample dataset, i.e., length, width, circumference, for manual and 3D scan results.
(2)$$\sigma =\sqrt{\frac{\sum_{i=1}^{n}{\left(x_{i}-\bar{x}\right)}^{2}}{n-1}}$$
where:

σ—standard deviation of the dataset.

To provide information about the fit of the acquired data, the Pearson correlation coefficient (Pearson’s r) (3) was used:(3)$$\;\rho_{x,\;y}=\frac{cov\left(X,\;Y\right)}{\sigma_{x}\sigma_{y}}$$
where:

ρ—Pearson’s r coefficient,

$\sigma_{x}$—standard deviation of the variable *X*, which represents one of the parameters from manual results, i.e., length, width or circumference,

$\sigma_{y}$—standard deviation of the variable *Y*, which represents one of the parameters of the 3D scan results, i.e., length, width or circumference,

cov—covariance.

Then, the quality of the model fit has been analyzed with the implementation of the R-squared (R^2^)coefficient of determination, using the following by Formula (4):(4)$$R^{2}=1-\frac{RSS}{TSS}=1-\frac{\sum_{i=1}^{n}{\left(y_{i}-\hat{y_{i}}\right)}^{2}}{\sum_{i=1}^{n}{\left(y_{i}-\bar{y}\right)}^{2}}$$
where:

RSS—residual sum of squares,

TSS—total sum of squares,

$y_{i}$—actual value for the current sample in the dataset,

$\hat{y_{i}}$—predicted value for the current sample in the dataset,

$\bar{y}$—mean of the y  values in the n-sample dataset.

Last but not least, the root mean squared error (RMSE) defined as the difference between manual and 3D scan results was calculated to estimate the absolute reliability of 3D scanner measurements (Formula (5)) has been applied:(5)$$RMSE=\sqrt{\overset{n}{\underset{i=1}{\sum }}\frac{{\left(\hat{y_{i}}-y_{i}\right)}^{2}}{n}}$$

Mean values with standard deviations (SD) for the aforementioned anthropometric measurements obtained in both methods, along with the RMSE, Pearson’s r coefficient and *R*^2^ for differences between manual and 3D scan results are presented in Table 2.

Linear relationships between manual and 3D scan results for three key hand anthropometric data with R^2^ coefficients are presented in Figure 4.

In this study, the dimensional allowances (DAs) were calculated as the differences between the bare and gloved hand for three anthropometric parameters, i.e., t length, width and circumference. For this purpose, the aforementioned dimensions for the bare and gloved hand were measured using the selected software, according to the landmarks positions specified in Table 1. The mean numerical values of performed measurements with SD are presented in Table 3.

Absolute and relative DAs were calculated based on the gathered data of the bare and gloved hand. For linear dimensions, i.e., length and width, the absolute DA was defined as a difference between the subject with and without the protective glove according to the Formula (6). For the circumference, Cloud-to-Cloud distance computation between the bare and gloved hand with the use of CloudCompare software was applied. Relative DAs were considered in this study to understand the relation of changes between the bare and gloved hand dimensions. Formulas (6) and (7) describe the way of calculating relative DA for linear dimensions and circumference for each sample in the dataset:(6)$$\delta_{z}=\frac{absolute\;DA}{z}\cdot 100\%=\frac{z^{\prime }-z}{z}\cdot 100\%$$
where:

$\delta_{z}$—relative DA [%],

z—dimension (length L/width W) corresponding to the position of the bare hand in the glove for the current sample in the dataset,

$z^{\prime }$—dimension (length $L^{\prime }$/width $W^{\prime }$) of the hand in the protective glove for the current sample in the dataset.

In each sample, to calculate the relative DA for the circumference, maximum distance obtained from the software was used and implemented in Formula (7):(7)$$\delta_{C}=\frac{max\;Cloud-to-Cloud}{CC}$$

The absolute DA with the minimum, maximum and mean values with corresponding SD as well as the mean values of relative DAs obtained according to the operations described above are presented in Table 4.

The key anthropometric results obtained from 3D scans analysis for three examined parameters: length, width and circumference of the bare and gloved hand are presented in Figure 5.

For linear dimensions, such as length and width, mean of absolute and relative differences between the bare and gloved hand with corresponding SD values are presented in Figure 6. In turn, for the circumference, mean of the maximum dimensional allowances are highlighted along with relative Das, related to the fact that the main interest of this study is the maximum dimensional allowances for the selected anthropometric parameters.

## 4. Discussion

Most importantly, the results of quality validation and 3D scanner accuracy and repeatability demonstrate that Artec Eva 3D scanner measurements of hand length, width and circumference are in accordance with the proposed methodology and as accurate as manual measurements. The proposed application of landmarks and a positioning base for the hand considerably improved the dimensioning process and the quality of results. This makes it possible to precisely match and superimpose scans of a bare and gloved hand. Pearson’s *r* correlation coefficient may be used to measure the strength of the linear relationship between manual measurements and those performed using an Artec Eva 3D scanner. A comparison of the two methods of measurement yielded correlation coefficients equal to 0.94 for length, 0.92 for width and 0.86 for circumference, respectively, which signifies a high and positive correlation. The values of *R*^2^ presented along with the linear relationship between 3D scans and manual results (see Figure 4) show a satisfactory model fit with points close to the linear trend line, and are equal to 0.89, 0.85 and 0.73 for the length, width and circumference, respectively. These results are also supported by previous literature that 3D scanners and manual measurements give consistent results for documenting bullet trajectories and the location and size of injuries during autopsies [24] and hand circumference [3]. Thus, a handheld 3D scanner is reliable instrument for measuring 3D hand anthropometric data and can offer as much more measurement information about crucial hand anthropometric data such as length, width and circumference as a traditional dimensioning method. Moreover, because the results of 3D scanning can be accurate to the micron, it should be noted that the tape measure used in this research is less accurate than the software and could decrease data fit correlation as well as increase errors between the two data sets.

The novelty of this study consisted of the fact that landmarks were used for determining the hand position in the glove and as well as for the superimposing and dimensioning procedures. This study found that as compared with the hand circumference obtained by the 3D scanner and manual measurements, the RMSE for length and width measurements was 0.41 mm which means the lowest differences, whereas for the hand circumference it was equal to 0.69 mm. The RMSE was used to understand the measurement discrepancies between the two methods presented in Table 2. It shows small differences in measurements of hand length, width and circumference between the two methods. Analysis of three different coefficients such as Pearson’s r, R2 and RMSE, whether for 3D scanning or manual measurement, circumference dimensioning is less accurate than length and width (linear dimensioning), which is most likely due to the following reasons. Firstly, the anatomical structure and curvature of the hand circumference are more complex than its length and width [4]. Secondly, as we discussed before, hand posture assessment is a well-known challenging problem. The identification of the position of the hand is very troublesome because adjusted slight change in its posture [13], can have a larger impact on circumference than on the width or.

Table 3 presents the numerical values of measurements taken from 3D scans for three key hand anthropometric data: length, width and circumference for the hand with and without the protective glove, with mean and corresponding standard deviation values. Data contained therein, along with their visual interpretation presented in Figure 5, reveal the significant changes in dimensions during the wearing of protective gloves. The region of the changes is shown as a difference between the two plotted lines used for the bare and gloved hand. This leads us to take a close-up look at the dimensional allowances whose absolute and relative values were calculated and are presented in Table 4.

The highest observed absolute DAs were 19.42 mm for length, 15.70 ± 2.43 mm for circumference and 13.5 mm for. As regards the maximum dimensional allowance for circumference, the mean of maximum values calculated for each trial in the software was used. Figure 6 shows the mean values of the obtained data with error bars corresponding to the standard deviations (SD) visualizing the highest absolute dimensional allowances (DA) for length (16.90 ± 1.91 mm), circumference (15.70 ± 2.43 mm) and width (12.00 ± 1.49 mm).

According to the relative differences between thehand with and without the protective glove, the highest values are equal to (13.77 ± 1.91%) for width, (7.96 ± 1.25%) for circumference, and length (5.90 ± 0.67%). This level of dimensional allowances (DA) could have a significant impact on the safe interaction between the subject and environment during the wearing of PPE, causing discomfort and limiting access to the tools, control panels, etc., as discussed precisely in the Introduction.

## 5. Conclusions

This study focuses on a novel approach to acquiring key anthropometric data for the bare and gloved hand from 3D scans, along with a dimensioning procedure for determining absolute and relative protective glove dimensional allowances, which may impair the user’s access to elements of the working environment.

In this study, three key anthropometric parameters, i.e., the length, width and circumference of the bare and gloved hand, were measured manually and with the use of software analyzing 3D scans.

The study was divided into two parts. The first one focused on the accuracy of the 3D scanner and the quality validation of the obtained results. To this end, we compared three parameters of the bare hand, i.e., length, width and circumference, taken manually and obtained from 3D scan. In the second step, bare and gloved hand dimensions extracted from 3D scans were compared in terms of length, width and circumference to determine dimensional allowances. In this step, two different approaches to measurements (linear dimensioning and cloud superimposing) were used.

Preliminary results confirmed that length, width and circumference measurements obtained from 3D scans are just as accurate as manual ones. Thus, the main results subsequently served as input in Cloud-to-Cloud computation and the size of the bare hand was subtracted from that of the gloved hand to determine absolute and relative glove dimensional allowances.

The method of hand positioning and the proposed set of supporting dimensioning landmarks constitute the main novelty of this paper. The palm rest surface used as a base for fixing the hand in the same position during the scanning and re-scanning procedures did not affect dimensioning limitations and substantially helped control the natural tendency of the hand to change its position, as discussed in the Introduction. On the other hand, landmarks determining the hand’s position in the glove were limited to the minimum, considerably simplifying the superimposing and dimensioning procedures.

Based on 3D scans, it was noted that the glove did not perfectly fit the subject’s hand, as indicated by the distance between the tips of the glove fingers and the human digits resulting from different positions of the bare hand and the hand in the protective glove. However, the focus of this study was the measurement method. For this reason, we selected a glove for circumferential rather than the longitudinal fit of the hand (the latter was not relevant in this study). Moreover, the positioning method involving a palm rest surface used for fixing the hand was not found to be suitable for dimensioning the digits due to a problem with their correct positioning during the scanning and re-scanning procedures. The problem needs to be addressed in future experiments.

The determined dimensional allowances amount to approx. 10%, limiting the glove user’s access to tools, control panels, and other elements of the work environment with which they interacts. The obtained results are useful for designers, especially in terms of designing keys on control panels and LCD touch displays and monitors integrated with machines.

## Figures and Tables

**Figure 1 ijerph-20-02645-f001:**
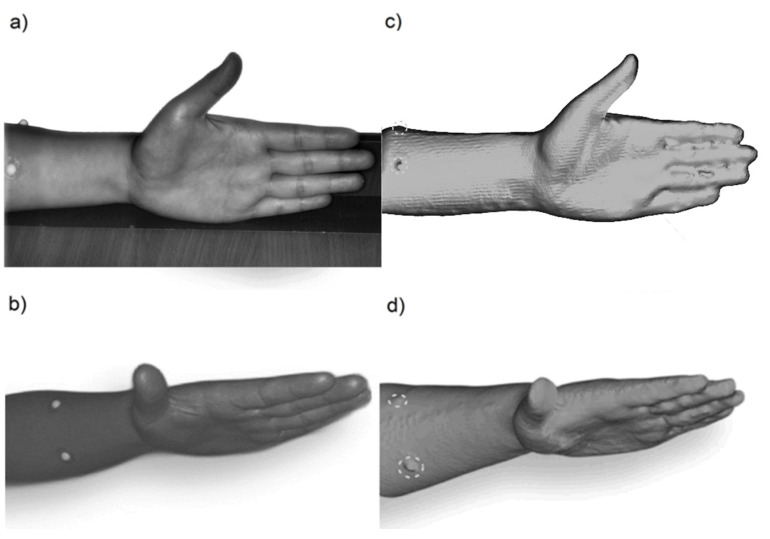
Bare hand with the palm rest surface used as a base for fixing the hand in the same position during the scanning and re-scanning procedure with landmarks supporting hand dimensioning (**a**) along with its image during the 3D scanning procedure (**a**,**b**) and corresponding 3D scans (**c**,**d**) with highlighted landmarks.

**Figure 2 ijerph-20-02645-f002:**
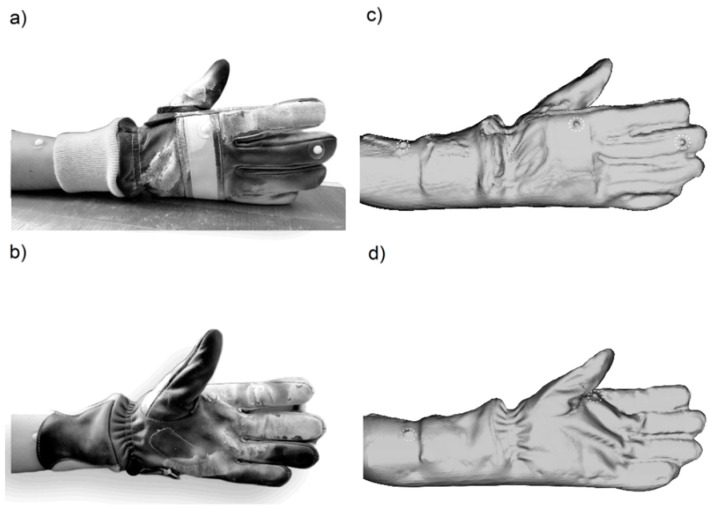
Hand in the protective glove with the palm rest surface used as a base for fixing the hand in the same position during the scanning and re-scanning procedure with landmarks supporting hand dimensioning (**a**) along with its image during the 3D scanning procedure (**a**,**b**) and corresponding 3D scans (**c**,**d**) with highlighted landmarks.

**Figure 3 ijerph-20-02645-f003:**
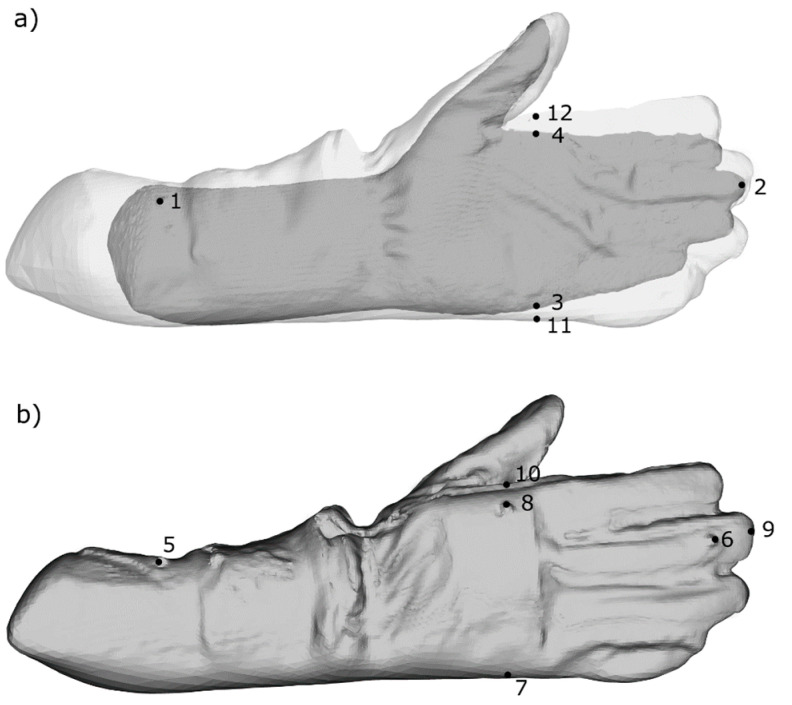
Landmarks reference points for dimensional allowances calculation: (**a**) the bare hand with the landmarks used for determining its dimensions (linear dimensioning) and superimposed clouds of the bare and gloved hand for circumference dimensioning; (**b**) the glove with the placed landmarks used to define the dimensions of the glove and the position of the hand therein (linear dimensioning).

**Figure 4 ijerph-20-02645-f004:**
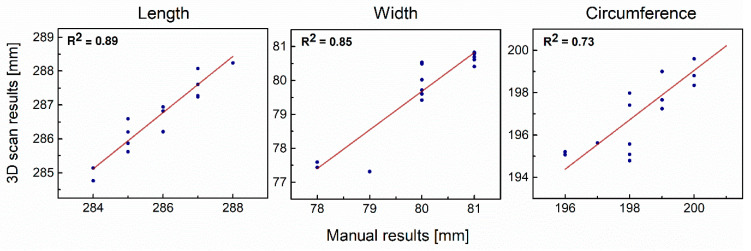
Linear relationship between manual and 3D scan results for length, width and circumference.

**Figure 5 ijerph-20-02645-f005:**
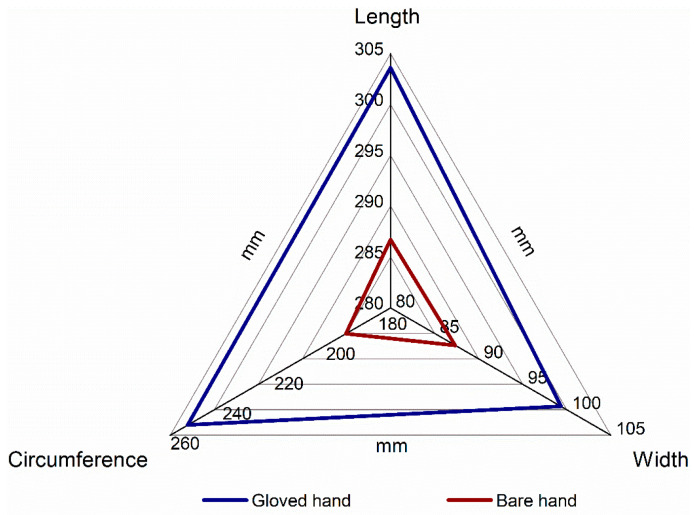
Spider chart of the key anthropometric parameters obtained from 3D scan analysis for the bare and gloved hand.

**Figure 6 ijerph-20-02645-f006:**
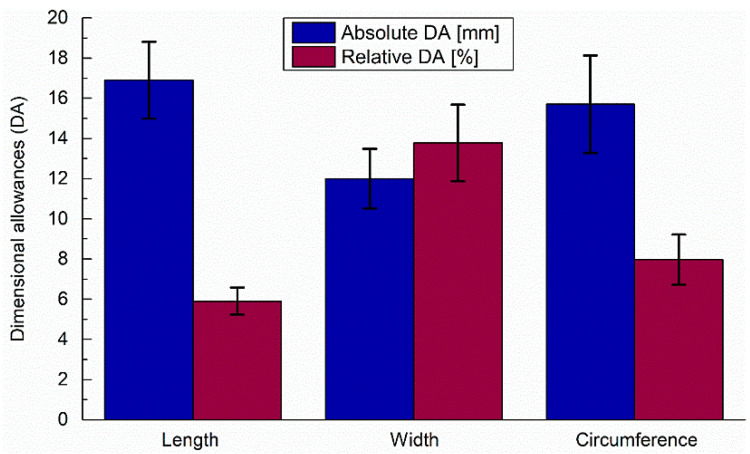
Mean of the absolute and relative dimensional allowances for the three hand anthropometric data between the bare and gloved hand: length, width and circumference with error bars representing the standard deviation of measurements.

**Table 1 ijerph-20-02645-t001:** The description of the linear and circumference measurements with reference points for 3D scans analysis.

	Measurement	Description	Symbol and Landmarks	Figure No.
**Bare hand**	Length	Distance on the palmar surface from the landmark placed on the forearm to the tip of the 3rd digit	L0: 1–2	Figure 3a
	Width	Width measured 20 mm from the crotch between thumb and index finger	W0: 3–4	
	Circumference	Circumference measured 20 mm above the thumb base	CC: Across 3–4	
**Gloved hand**	Length	Distance on the dorsal glove surface, from the landmark placed on the forearm to the landmark corresponding to the tip of the 3rd digit of the hand	L: 5–6	Figure 3b
	Width	Distance on the dorsal glove surface, from the rest surface to the landmark corresponding to the top hand surface	W: 7–8	
**Glove**	Length	Distance on the dorsal side of the glove surface, from the landmark placed on the forearm to the tip of the 3rd digit of the glove	L’: 5–9	Figure 3b
	Width	Distance on the dorsal glove surface, from the rest surface to the top surface of the glove	W’: 7–10	
	Circumference	Circumference measured, 20 mm above the thumb base of the hand in the glove	CC’: Across 11–12	Figure 3a

**Table 2 ijerph-20-02645-t002:** Mean values of measurements for the bare hand obtained from manual and 3D scan analysis with corresponding standard deviations (SD), root mean squared errors (RMSE), as well as Pearson’s *r* and *R*-square (*R*^2^) values for differences between manual and 3D scans results.

Measurement	Manual [mm]	3D Scan [mm]	RMSE	Pearson’s *r*	*R* ^2^
Length	285.93 ± 1.22	286.71 ± 1.07	0.41	0.94	0.89
Width	80.07 ± 1.03	79.75 ± 1.28	0.41	0.92	0.85
Circumference	198.47 ± 1.46	197.25 ± 1.99	0.69	0.86	0.73

**Table 3 ijerph-20-02645-t003:** Mean values from 3D scan analysis with corresponding standard deviation for the bare and gloved hand.

Measurement	Hand [mm]	Glove [mm]
Length	286.71 ± 1.07	303.61 ± 2.20
Width	87.35 ± 1.78	99.35 ± 1.10
Circumference	197.25 ± 1.99	258.28 ± 3.07

**Table 4 ijerph-20-02645-t004:** Mean, minimun and maximum values of absolute and relative dimensional allowances (DAs) for gloved hand for three key hand anthropometric data: length, width and circumference.

Dimensional Allowances	Absolute [mm]			Relative [%]
	Mean	Min	Max	
Length DA	16.90 ± 1.91	13.59	19.42	5.90 ± 0.67
Width DA	12.00 ± 1.49	8.8	13.5	13.77 ± 1.91
Circumference DA	7.30 ± 2.02	0.30 ± 0.54	15.70 ± 2.43	7.96 ± 1.25

## Data Availability

The data presented in this study are available on request from the corresponding author.

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
