# Peer review of "3D Hand Scanning Methodology for Determining Protective Glove Dimensional Allowances"

_ijerph, 2023, doi:10.3390/ijerph20032645_

Round 1
Reviewer 1 Report
1. The object of the research was a woman, 25 years old (page 3). Is anthropometrics convincing using single subject research? This is the major problem.
2. Firefighter has unique anthropometric data, which published in recently issue of journal. Only one woman and unclear occupational description are questionable. Moreover, most of the firefighter are males.
3. What kind of scenarios are protective gloves used in? In the introduction section, there is a lack of information on the importance of studying protective gloves and the associated safety hazards, as well as a lack of detail on the dimensional difference. In the materials and methods of this paper, there is no mention of the criteria for selecting gloves for experiments and relevant references.
4. The hand dimension definition of Table 1. are not clear. Dimension labeling procedure should be added.
5. The Discussion section of the study should be revised entirety. Firstly, the necessity of Figs. 5, 6 is questionable and, in this paper, Figs. 5, 6, are not explained in depth. Furthermore, there is no discussion around the measurement method, which is the major focus of this paper.
6. The performance of the three algorithms in this study may lead to problems with the screening and regularization of the step threshold factors.
7. Page 10 The discussion section, especially the last paragraph, is poorly written, very broad and directs the questions to the introduction section, which is not up to the standard of publication. It is recommended to organize the introduction section and make a discussion based on the main results of the paper.
8. The authors noted that the focus of this paper is on innovation in measurement methods. However, there is no validation of measurement methods. such as the intraclass correlation coefficient (ICC). In addition, the research focus of this paper is not particularly strong enough in the methods and discussion section.
9. The English editing is strongly recommended.
Reviewer 2 Report
The manuscript presents an interesting study on 3D scanning and modelling of body parts by means of a novel supporting dimensioning landmarks set.
The manuscript is well organized and can be of interest for the journal’s target audience. However, before considering it for publication, some major improvements are needed.
First, in the introduction, the research motivations should be expanded to better highlight how the current study can augment knowledge in the industrial ergonomics field.
In section 2, a flowchart or a table illustrating the proposed research methodology should be provided, indicating for each phase of the proposed procedure inputs, outputs, and tools.
Another criticality is represented by the discussion of results. On the one hand, section 4 examines the case study results; on the other hand it fails in providing information about the practical implications of the study, the major research findings, and its limitations.
Additionally, some minor issues concern:
In the introduction it is said that "Paper [7] presents the analysis…”. It would be better to say: "Griffin et al. [7] present the analysis…”
The title does not represent the contents of the manuscript.
Reviewer 3 Report
Authors have done nice work. However the comments could be addressed.
1. How the anthropometric variables of the hand were measured?
2. There are three types of Anthropometic Data (Structural, Functional, Newtonian) commonly used. Which one you have used for this study? Justify.
3. Standard measurements can be extracted using parametric or non parametric. Which method have you followed-Justify.
4. Why you used the spider chart of the key anthropometric parameters (Figure 5). Comparing values across a circle is harder than comparing them on a straight line. Justify.
5. Add additional references published in 2022.
Round 2
Reviewer 1 Report
I have no more questions.
Reviewer 2 Report
The Authors have sufficiently improved the manuscript. Hence, it can be considered for publication.
Reviewer 3 Report
Authors have addressed all the comments